# The Composites of PCL and Tetranuclear Titanium(IV)-oxo Complexes as Materials Exhibiting the Photocatalytic and the Antimicrobial Activity

**DOI:** 10.3390/ijms22137021

**Published:** 2021-06-29

**Authors:** Barbara Kubiak, Aleksandra Radtke, Adrian Topolski, Grzegorz Wrzeszcz, Patrycja Golińska, Ewelina Kaszkowiak, Michał Sobota, Jakub Włodarczyk, Mateusz Stojko, Piotr Piszczek

**Affiliations:** 1Faculty of Chemistry, Nicolaus Copernicus University in Toruń, Gagarina 7, 87-100 Toruń, Poland; basiak0809@gmail.com (B.K.); topolski@umk.pl (A.T.); wrzeszcz@umk.pl (G.W.); kaszkowiake@gmail.com (E.K.); 2Faculty of Biological and Veterinary Sciences, Nicolaus Copernicus University, Lwowska 1, 87-100 Toruń, Poland; golinska@umk.pl; 3Centre of Polymer and Carbon Materials, Polish Academy of Sciences, M. Curie-Skłodowskiej 34, 41-819 Zabrze, Poland; msobota@cmpw-pan.edu.pl (M.S.); jwlodarczyk@cmpw-pan.edu.pl (J.W.); mstojko@cmpw-pan.edu.pl (M.S.)

**Keywords:** PCL, titanium(IV)-oxo complex, composite, physicochemical properties, photocatalytic activity, reactive oxygen species, antimicrobial activity

## Abstract

Excessive misuse of antibiotics and antimicrobials has led to a spread of microorganisms resistant to most currently used agents. The resulting global threats has driven the search for new materials with optimal antimicrobial activity and their application in various areas of our lives. In our research, we focused on the formation of composite materials produced by the dispersion of titanium(IV)-oxo complexes (TOCs) in poly(ε-caprolactone) (PCL) matrix, which exhibit optimal antimicrobial activity. TOCs, of the general formula [Ti_4_O_2_(O^i^Bu)_10_(O_2_CR’)_2_] (R’ = PhNH_2_ (**1**), C_13_H_9_ (**2**)) were synthesized as a result of the direct reaction of titanium(IV) isobutoxide and 4-aminobenzoic acid or 9-fluorenecarboxylic acid. The microcrystalline powders of (**1**) and (**2**), whose structures were confirmed by infrared (IR) and Raman spectroscopy, were dispersed in PCL matrixes. In this way, the composites PCL + *n*TOCs (*n* = 5 and 20 wt.%) were produced. The structure and physicochemical properties were determined on the basis of Raman microscopy, thermogravimetric analysis (TGA), differential scanning calorimetry (DSC), electron paramagnetic resonance spectroscopy (EPR), and UV–Vis diffuse reflectance spectroscopy (DRS). The degree of TOCs distribution in the polymer matrix was monitored by scanning electron microscopy (SEM). The addition of TOCs micro grains into the PCL matrix only slightly changed the thermal and mechanical properties of the composite compared to the pure PCL. Among the investigated PCL + TOCs systems, promising antibacterial properties were confirmed for samples of PCL + *n*(**2**) (*n* = 5, 20 wt.%) composites, which simultaneously revealed the best photocatalytic activity in the visible range.

## 1. Introduction

The interest in titanium(IV)-oxo complexes (TOCs) is inter alia due to their physicochemical and optical properties (e.g., luminescent, photochromic, and photocatalytic properties), which can be compared to titanium dioxide particles (TiO_2_) [1,2,3,4,5]. From a practical point of view, photocatalytic activity of TOCs is important; for example, their application in the photodegradation of organic water pollutants or the formation of self-cleaning materials [6,7,8,9,10]. However, the systems mentioned above differ in the mechanism responsible for the course of photocatalytic processes. Titanium dioxide is a cheap, non-toxic semiconductor of high-value bandgap energy (*E*_g_ of rutile and anatase, the two most known polymorphic forms of titania are equal to 3.02 eV, and 3.20 eV, respectively), which corresponds to the radiation energy in the UV range [11]. The characteristic feature of semiconductor nanoparticles (SNP) is their ability to transfer electrons from the valence band to the conduction band as a result of photon absorption, which leads to the formation of the electron (*e*-) and hole (*h*+) pair (exciton) [12,13]. Excitons exhibit strong redox properties and can react with water and oxygen molecules adsorbed on the SNP surface and lead to the generation of reactive oxygen species (ROS) [14,15]. The following factors influence the photocatalytic activity of TiO_2_-based nanomaterials: particle size, surface morphology, structure, radiation intensity, and exposure time [15,16]. Friehs et al. drew attention to the fact that the photoinduced toxicity of TiO_2_ P25 powder is stronger than other TiO_2_ nanoparticles. Moreover, the rutile form revealed lower toxicity than the anatase particles [11]. The presence of organic ligands stabilizing the {Ti_a_O_b_} cores in the structures of oxo-complexes can work as a photosensitizer and lead to the absorption range widening towards the visible light [16,17,18]. Due to this effect, the electrons transfer distance decrease between HOMO–LUMO orbitals of the ligands and the {Ti_a_O_b_} core, and the energy of the absorption band is lower [16,19,20]. Analysis of the literature data shows that photocatalytic activity of TOCs can be controlled by way of ligand functionalization, especially of carboxylate groups [21]. The results of our previous research on the photocatalytic activity estimation of composites, which contained PMMA and TOCs consisting of {Ti_4_O_2_} cores, confirmed this [17,18]. According to these reports, systems stabilized with 9-fluorenocarboxylate and 4-aminobenzonate ligands exhibited the best properties. The results obtained r were so promising that we decided to continue the work on the above-mentioned TOCs group.

In searching for new, more effective agents/materials with bactericidal effect, attention has been drawn to metal oxides exhibiting photocatalytic properties [22,23]. The direct contact of the microorganism with the photocatalyst surface leads to the damage of the integral cell membrane due to the action of the ROS, mainly hydroxyl radicals (•OH) and hydrogen peroxide (H_2_O_2_) [24,25]. According to Foster et al., in the first stage, the bacterial cell contents leak, followed by cell lysis, up to the full mineralization of the microorganism [24,26]. Photocatalytic activity of TOCs suggests that also, in this case, that ROS can be generated on their surface. The hydrophobicity of TOCs grains/powders and their sensitivity on hydrolysis processes caused the previous interest of this group of compounds as potential antimicrobial agents to be limited. Although studies of the hydrolyzed form of the complexes consisted of {Ti_4_O_2_} cores stabilized by two triclosan ligands, they revealed antibacterial activity against *Staphylococcus aureus* [27]. However, it should be noted that the triclosan ligand, which was used for oxo-complex stabilization, is a potent antibacterial and antifungal agent [28]. Therefore, in this case, the presence of the triclosan (as a hydrolysis product) can be seen as a main bactericidal factor. The results of our earlier investigations on [Ti_4_O_2_(O^i^Bu)_10_(O_2_CR’)_2_] and [Ti_3_O(O^i^Pr)_8_(O2CR’)_2_] (R’ = C_13_H_9_, PhCl, PhNO_2_, PhNH_2_, PhOH) complexes, dispersed in PMMA matrix, proved their resistance on the hydrolysis processes and made it possible to evaluate their photocatalytic activity [17,18,29,30]. Moreover, studies of (PMMA + TOCs) composites (TOCs = [Ti_3_O(O^i^Pr)_8_(O_2_CPhR’)_2_] and [Ti_4_O_2_(O^i^Bu)_10_(O_2_CPhR’)_2_] (R’ = NH_2_ and OH)) confirmed their phototoxicity against *Staphylococcus aureus*, *Escherichia coli*, and *Candida albicans*, their effectiveness was 85–99% [18]. According to earlier reports, it should be noted that in order to produce a composite material with suitable microbicidal properties, the TOCs of a specific core structure {Ti_a_O_b_} and/or functionalized carboxylate ligands must be introduced to the polymer matrix. 

In our studies, we have focused on studying the microbicidal activity of tetranuclear titanium(IV)-oxo complexes with high photocatalytic activity [17,18]. Synthesized compounds were dispersed in the poly(ε-caprolactone) (PCL) matrix. PCL is a biodegradable (degradation time from 2 to 3 years) thermoplastic polymer, which is increasingly used in the production of medical devices, e.g., catchers, stents, intrauterine devices [31,32,33,34,35,36,37], as well as environmentally friendly packaging for food products [38]. The analysis of the previous reports shows the growing interest in modifying PCL films/coatings to obtain biodegradable materials, which can be antimicrobial agent release carriers, e.g., antimicrobial peptides [39] or antibiotics [37]. Moreover, work is underway on composite materials with appropriate antimicrobial activity, which are obtained by introducing an inorganic agent, such as ZnO and CuO, into the PCL matrix [40]. The use of the PCL/TiO_2_ system is especially interesting in the production of composite fibers using the electrospinning method [41,42,43,44,45,46]. It was noticed that the addition of TiO_2_ particles to PCL matrix influences the reduction of its water absorption and improves the mechanical properties (e.g., compressive strength) [41]. Moreover, Kiran et al., investigating the antimicrobial activity of PCL/TiO_2_ coatings and irradiated by UV light produced on the titanium implants surface, exhibited improvement of their biological properties and antibacterial activity [42]. For this reason, in our investigations, we have focused on PCL + TOCs systems (TOCs = [Ti_4_O_2_(O^i^Bu)_10_(O_2_CR’)_2_]) containing R’ = 9-fluorenocarboxylic groups and 4-aminobenzoic ones as the stabilized ligands. The novelty of the research carried out was the use of the PCL + *n*TOCs system (*n* = 5 and 20 wt.% TOCs) to obtain extruded fittings, which reveal the antimicrobial properties. The research results could be used for the production of composite coatings with self-disinfecting surfaces.

## 2. Results

The TOCs, of the general formula [Ti_4_O_2_(O^i^Bu)_10_(O_2_CR’)_2_] (R’ = PhNH_2_(**1**) and C_13_H_9_ (**2**)), were isolated from a 4:1 molar ratio mother liquor of Ti(O^i^Bu)_4_ and organic acids (HO_2_CR’; R’ = C_13_H_9_, PhNH_2_) according to the procedure described earlier [17,30]. Analysis of spectra registered using diffuse reflectance infrared Fourier transformation (DRIFT) and Raman spectroscopy allowed for their structure to be confirmed [17,30]. SEM images of (1) and (2) oxo-complexes micro-powders isolated from the mother liquor are presented in Figure 1 (moreover, the SEM EDX data concerning (1) and (2) are shown in Appendix A).

### 2.1. The Production of Polymer + TOCs Composites and Studies of Their Physicochemical and Thermal Properties

The produced (**1**) and (**2**) microcrystalline powders were dispersed in the poly(ε-caprolactone) (PCL) matrix and then, applying the injection molding method, produced the fittings containing *n* = 5 and 20 wt.% of TOCs (PCL + *n*(TOCs composite). The size of TOCs grains and their distribution in the produced fittings are shown in Figure 1. Analysis of SEM images exhibited the relatively homogeneous distribution of TOCs grains in the PCL matrix. In samples that contain the 5 wt.% TOCs, grains of sizes from 3 to 15 μm predominated, however, larger grains (size 30–90 μm) were also found. The increasing of TOCs content up to 20 wt.% resulted in a uniform distribution of larger grains (ca. 20–70 μm) in the entire volume of the fittings, regardless of the oxo-complex type (Figure 1 and Appendix A).

The presence of TOCs in all studied samples and their structural stability during the fabrication of composite fittings has been confirmed by the registration of Raman microscope maps (Figure 2 and Figure 3). The structural stability confirmation of (**1**) and (**2**) was especially important, as the sample was heated up to 70 °C during injection molding, which could change their structure in dispersed compounds. For comparison, the Raman spectra of (**1**) and (**2**) dispersed in PMMA at room temperature are presented.

Moreover, the presence of TOCs grains in the PCL matrix was confirmed by the SEM EDX method (Table 1, Figure 4). Analysis of these data confirms the presence of peaks assigned to Ti, O, and C in EDX patterns of the samples containing TOCs micro grains, while in the case of the pure PCL sample, the Ti peaks were not found (the Al peak is derived from microscope equipment).

Possible changes in thermal properties of the studied composites, caused by the addition of TOCs, were estimated using thermogravimetric analysis (TGA) and differential scanning calorimetry (DSC). The measurements were carried out in the temperature range 30–550 °C, at nitrogen atmosphere, and obtained results are presented in Table 2 and Figure 5a,c). Thermal decomposition of PCL proceeded in one stage between 20 and 460 °C (Figure 5a). In general, the dispersion of (**1**) and (**2**) grains in PCL matrix slightly reduces the decomposition temperature of most investigated composites (2–5 °C), which can be associated with the decomposition of TOCs below 320 °C (in thermograms, the weak decomposition stage below 320 °C was found (Figure 5a). The exception was the PCL + 20(**2**), for which the decomposition temperature was decreased and their one-stage thermal decomposition proceeded between 260 and 420 °C. For PMMA composites, it can be seen that the decomposition process takes place in two stages, as described in the publication [18]. For comparison, TGA studies of PMMA and PMMA + 20TOCs (TOCs = (**1**) and (**2**)) samples revealed that addition TOCs grains decreased the first stage of thermolysis temperature and increased the second one (Figure 5c, Table 2).

The results of DSC thermal analysis of pure polymers (PCL, PMMA) and composites containing (**1**) and (**2**) powders are presented in Figure 5b,d and Table 2. The thermogram of pure PCL showed a single endothermic peak at around 66 °C (Tm), which was attributed to the polymer melting. Since DSC measurements were carried out between 30 and 550 °C, the glass transition temperature (T_g_) of PCL (T_g_ = −61 °C [44]) could not be observed. The dispersion of TOCs grains in the PCL matrix practically does not influence the melting temperature (Table 2). The second endothermic peak at ca. T_d/max_ = 402 °C, which was found in the PCL thermogram was assigned to the polymer decomposition (the polymer thermal decomposition starts at 344.9 °C). In the case of composites containing 5 wt.% of TOCs grains, a slight reduction of T_d/max_, i.e., 4 °C for PCL + 5(**1**) and 6 °C PCL + 5(**2**) was observed (Table 2). The beginning of the decomposition was registered at ca. 342 °C (**1**) and 354 °C (**2**), respectively (Table 2). In the thermogram of PCL + 20(**1**) sample, the additional weak peak was found at T_d/max_ = 332 °C (Figure 5b). This transition can result from the thermal decomposition of (**1**), whose start was found at c.a 290 °C. Endo–exo transitions, which were noticed at 328 and 374 °C, respectively in the thermogram of PCL + 20(**2**), were also attributed to composite decomposition (Table 2, Figure 5b). Analysis of DSC thermograms of PMMA and PMMA + 20TOCs (TOCs = (**1**) and (**2**)) showed that the addition of TOCs eliminated the thermal effect occurring at ca. 151 °C for pure PMMA (Table 2, Figure 5b) [18]. Furthermore, in these cases, exo–endo transitions between 200 and 440 °C, resulted from the composite thermal decomposition, however, the increase of T_d_ of both composites versus the pure PMMA was observed (Table 2, Figure 5d).

The mechanical strength studies of the polymer and composite fittings subjected to the static tensile test allowed us to determine the effect of introducing micro grains of complexes (**1**) and (**2**) into the PCL matrix. Samples containing 20 wt.% TOCs were subjected to these measurements, and obtained results are presented in Table 3. In general, it should be noted that the introduction TOCs grains to the PCL matrix increased the Young’s modulus value (E) in comparison to pure PCL, wherein this increase was greater after the introduction of (**1**) than (**2**). Simultaneously, tensile strength and strain at break decreased their values independently according to the type of added oxo-complex (Table 3). In the case of the compressive test of the samples, the analogous change in Young’s modulus value (E) of the composite compared to pure PCL was observed, but the differences were not visible. These results show that the elastomeric character of the samples was kept even with 20 wt.% of addition TOCs grains.

### 2.2. Estimation of Photocatalytic Activity of the Oxo-Complexes

The UV–Vis–DRS spectra of produced PCL + TOCs composites were recorded to determine of the PCL absorption maximum position before and after the introduction of 5 and 20 wt.% (**1**) and (**2**) (Figure 6). Analysis of these data revealed the clear shifting of absorption maximum towards visible range from 236 nm for PCL up to 375–400 nm for produced composites. Therefore, in all our photocatalytic experiments samples were irradiated by visible light (λ = 350–2200 nm).

The photocatalytic activity was estimated on the basis of methylene blue solution (MB) decolorization during irradiation with visible light by 30 h. Changes in MB concentration in the presence of PCL + *n*TOCs (TOCs = (**1**) or (**2**)) composites are presented in Figure 7 and Table 4. The results of these investigations were compared with the studies of photocatalytic activity of PMMA + 20TOCs (TOCs = (**1**) or (**2**)) also irradiated by visible light (these studies were not carried out in our earlier works). According to the data presented in Figure 7a,b and Table 4, it should be noted that a 5 wt.% addition of the oxo-complexes (TOCs = (**1**) or (**2**)) to the PCL matrix increases photocatalytic activity of PCL + 5TOCs in comparison to the pure polymer, especially in the case of PCL + 5(**2**) (the reference rate constants were 00029 h^–1^ and 0.0078 h^–1^ for PCL + 5(**1**) and PCL + 5(**2**), respectively. According to presented data, a pure PCL showed no photocatalytic activity in the visible range; its activity was generally the same as for irradiated MB solution. The increase of TOCs content up to 20 wt.%, significantly improved the activity of studied materials, especially for the PCL + 20(**2**) system. Comparative studies carried out for the composites PMMA + 20TOCs (TOCs = (**1**) or (**2**)) revealed that photocatalytic activity of the PMMA + 20(**2**) composite was also clearly greater than the activity of (PMMA + 20(**1**) one (Figure 7c,d and Table 4).

### 2.3. Results of EPR Studies

EPR spectroscopy was used to detect paramagnetic species on the surface of the synthesized materials. Pure PCL polymer shows no EPR signal. However, paramagnetic centers were found in all samples of PCL + TOCs. Generally, the signals in the EPR spectra were very weak, especially for composites with a 5 wt.% admixture of oxo-titanium(IV) complexes. Therefore, only the EPR spectra of the samples with 20 wt.% addition of oxo-titanium(IV) complexes were analyzed in detail. Spectra registered for composites of PCL + 20(**1**) and PCL + 20(**2**) are presented in Figure 8. The EPR parameters and types of observed paramagnetic species (ROS) are summarized in Table 5.

Some EPR signals are marked with a question mark (Table 5 and Figure 8b) for two reasons. The first one (Table 5) concerns the value of the lowest g-factor for O_2_^–^ species, as the signals from both oxygen species overlapped. Another reason (Figure 8b) concerns the very weak and very broad signal for O^–^ species, which during accumulation disappeared probably due to slight fluctuations in the magnetic field.

### 2.4. Antimicrobial Activity of PCL + TOC Composites

Antibacterial and antifungal activities of PCL and PCL + *n*TOCs (*n* = 5 and 20 wt.%) are presented in Table 6. Generally, these studies revealed that enrichment of the polymer with TOCs grains decreased the number of microorganisms when compared with pure polymer (PCL). However, the biocidal effect of PCL + TOCs composites (R ≥ 2) was observed against bacteria, and when PCL was enriched with 5 wt.% TOCs containing 9-fluorenocarboxylic stabilized ligand (2), with one exception (*Staphylococcus aureus* ATCC 6538), or with 20 wt.% TOCs containing 4-aminobenzoic stabilize ligand (1) and 9-fluorenocarboxylic stabilize ligand (2). Gram-negative bacteria showed higher sensitivity to PCL + TOCs than Gram-positive bacteria, especially in the presence of 20 wt.% TOCs (2). In these cases, the reduction indexes were found to be between 3.5-4.5 and 2.0-2.6, respectively. In general should be noted the weak antimicrobial activity of studied samples against yeasts, i.e., Candida albicans. The PCL + 20(2) showed the best antifungal properties among studied systems (Table 6).

## 3. Discussion

The previous studies on films based on poly(methyl methacrylate) (PMMA) enriched with 20 wt.% TOCs (TOCs = [Ti_4_O_2_(O^i^Bu)_10_(O_2_CR’)_2_]; R’ = PhNH_2_ and PhOH) revealed their photocatalytic activity in the UVA/Visible range, and also, their clear bactericidal properties [17,18]. The studies were carried out on PMMA + TOCs composite films of 25–50 μm thickness, containing TOCs grains, with sizes ca. 3–5 μm (R’ = NH_2_) or ca. 100–300 μm (R’ = OH) [18]. Analysis of Raman spectra of the composites mentioned above confirmed structural stability of TOCs introduced to the polymer matrix and their low sensibility on hydrolysis processes. This prompted us to continue our research on the possibility of using multinuclear Ti(IV)-oxo complexes as an antimicrobial agent. This paper presents the research results on PCL + *n*TOCs (*n* = 5 and 20 wt.%) composite fittings of 1.5 mm thickness, produced with the use of equipment for plastics processing consisting of a twin-screw extruder and a pneumatic mini-injection molding machine. The tetranuclear oxo-complexes [Ti_4_O_2_(O^i^Bu)_10_(O_2_CR’)_2_] (R’ = PhNH_2_ (**1**) and C_13_H_9_ (**2**)) were used in the fabrication of test samples, which were irradiated with the visible light during all photocatalytic and microbiological experiments. Analysis of Raman microscopy data and SEM-EDX ones confirmed the uniform dispersion of TOCs grains (both (**1**) and (**2**)) in the whole volume of fittings and their structural stability during the production process. The size of dispersed TOCs grains depended on their content and, in the case of 5 wt.% amounted 3–90 μm while for 20 wt.% 20–70 μm. The TGA and DSC data analysis revealed that depending on the type of used polymer (PCL, PMMA), the thermolysis temperature was slightly lower for PCL + 20TOCs composites and increased for PMMA + 20TOCs ones, in comparison to the pure polymers. This effect can be related to the thermal decomposition of TOCs, for which T_dmax_ were ca. 310 and 291, 393 °C for (**1**) and (**2**), respectively (thermal decomposition of (**2**) proceeds in two stages), which dependently to decomposition temperature of the pure polymer (ca. 402 °C for PCL and ca. 365 °C for PMMA) can accelerate or decelerate composite thermal decomposition processes. 

The photocatalytic activity studies of PCL + *n*TOCs composites were preceded by the determination of their absorption maximum, which was carried out using the UV-Vis-DRS technique. Analyzing the registered data (Figure 6), it was noticed that the dispersion of TOCs grains into the PCL matrix results in a clear shift of their absorption maximum from the UV range (236 nm for pure polymer) towards the visible range (375–400 nm). Earlier studies of Ti(IV)-oxo complexes (**1**) and (**2**) showed that the bandgap magnitudes were 2.57 and 2.55 eV [17,18], respectively. The narrowing of the bandgap of TOCs compared to TiO_2_ (ca. 3.0–3.2 eV [11]) results from the photosensitizing effect of functionalized carboxylate ligands bonded to the {Ti_4_O_2_} core [47]. As a result, electrons excited under the influence of visible light can be transferred between the HOMO–LUMO of ligands with energy bands of the titanium-oxo core. Accordingly, the photocatalytic activity of studied PCL + *n*TOCs composites has been evaluated based on the MB decolorization for samples exposed to visible light. Analysis of data presented in Figure 7a,b, and Table 4 revealed the clear photocatalytic activity of PCL samples containing grains of TOCs (TOCs = (**1**) and (**2**)), in contrast to the pure PCL, which is inactive in this range of the light. The content of TOCs in the composite samples (*n* = 5 and 20 wt.%) and the type of carboxylic stabilization ligand (-O_2_CPhNH_2_ (**1**) and -O_2_CC_13_H_9_ (**2**)) are the main factors that influenced the photodecolorization of the MB solution. Moreover, the data provided in Table 4 suggests the possible relationship between the polymer matrix type and the composite photocatalytic activity. According to these data, the PMMA exhibits a weak ability to MB photodecolorization during exposure to visible light, unlike PCL, which is inactive. Comparing both studied composites, it should be noted that the photocatalytic activity of PMMA + 20TOCs composites was significantly lower than PCL + 20TOCs ones (Table 4).

Considering the relatively well-known mechanism of the photocatalytic action of TiO_2_ based materials, it can be assumed that after visible light photoexcitation of studied samples, the ROS can be generated on their surface [48]. Therefore, the microbiocidal activity estimation of the produced composites (containing 20 wt.% of TOCs) were preceded by an analysis of their EPR spectra, which allowed for the detection of ROS formation and their identification. In the EPR spectrum of PCL + 20(**1**), the characteristic anisotropic signals for O^−^ radical were found (Figure 8, Table 5). The formation of O^−^ type paramagnetic centers consists of removing the electron and stabilizing the radical formed by reduction of titanium(IV) to titanium(III) [49]. So, although the presence of titanium(III) is not excluded, the respective signals are not visible due to their much greater width compared to the signals of radicals and, consequently, their lower intensity. The EPR spectra of PCL + (**2**) are more complex and contain more lines. The literature data shows that the maximum g-factor (g_zz_) for O_2_^−^ is greater than for O^−^ species. [50]. On the other hand, titanium(III), due to a positive spin-orbit coupling constant, shows EPR signals at even lower g-values, below 1.99 [48,49,50,51,52]. This approach makes it possible to identify the presence of ROS of the type O_2_^−^ and O^−^ in PCL + (**2**) samples (Table 5). The above analysis does not exclude the presence of other oxygen-based radicals because the superoxide radical anion, O_2_^−^, the peroxy-type radicals (RCOO·) and O_2_^−^ of organic-type adducts could be indistinguishable by the CW-EPR spectroscopy [52]. The role of oxidative stress in titanium dioxide-induced antimicrobial activity was confirmed by several previously published reports [24,25,41,53,54]. The promote photo-oxidation and photo-reduction processes, which can lead to ROS generation, were also observed for TOCs systems [14,18,55]. The formation of such radicals as oxide (O^−^), superoxide (O_2_^−^) or hydroxyl (•OH), and non-radicals, e.g., hydrogen peroxide (H_2_O_2_) or hydroxyl (OH^−^) ions, has been confirmed [14,56]. The generation of ROS is considered as the main mechanism of antimicrobial activity. ROS are effective in vitro against Gram-positive, and Gram-negative organisms, including multidrug-resistant (MDR) isolates and fungi, as they cause several types of oxidative damage intracellularly. They are also highly active against viruses [57,58,59]; for example, H_2_O_2_ causes its antimicrobial action by a reaction with thiol groups in proteins, including enzymes, DNA, and bacterial cell membranes, while hydroxyl radical, which is a strong and nonselective oxidant, breaks DNA, peroxidates lipids, and carbonylates proteins [59,60]. Moreover, ROS oxidize the deoxyguanosine triphosphate (dGTP) and deoxycytidine triphosphate (dCTP) pools causing incorrect incorporation of bases into DNA. They also cause double-stranded breaks in DNA through disrupted repair intermediates [61,62,63]. The free radicals affect bacterial lipopolysaccharide (LPS) in outer membrane, peptidoglycan, and the phospholipid bilayer by causing peroxidation [54]. The relationship of photocatalytic activity and antimicrobial activity of PMMA + TOCs containing Ti_3_O or Ti_4_O_2_ cores (20 wt.%) and 4-hydroxybenzoic or 4-aminobenzoic ligands, against Gram-positive and Gram-negative bacteria, and yeasts was revealed in our previous studies [18]. All of the studied PMMA + TOC foils strongly (>99%) inhibited the growth of tested bacteria (*E. coli* and *S. aureus*). Both PMMA + TOCs with Ti_4_O_2_ core and 4-hydroxybenzoic or 4-aminobenzoic ligands also significantly inhibited the growth of yeasts of *C. albicans* not the complex of PMMA with a {Ti_3_O} core and stabilized by the -O_2_C-4-PhNH_2_ ligand [18]. In the presented paper, the antimicrobial activity of PCL + *n*TOCs (*n* = 5 and 20 wt.%., TOCs = (**1**) and (**2**)) composite samples were assessed, which were activated by visible light. Strong antibacterial activity for PCL containing 20 wt.% of the TOCs was found, like during the previous studies. However, in the case of 5 wt.% content of (**2**) in the composite sample such activity was recorded.

When the bactericidal activity of TOCs is studied, attention should be paid to the role of photocatalytically assisted antimicrobial processes activated by light. The importance of UV-assisted TiO_2_ photocatalysis for the inactivation of bacteria has been widely described in the existing literature [54,64,65,66]. However, in the case of PCL + *n*TOCs composites, the absorption maximum was shifted towards the visible range, which suggests that visible light plays a decisive role in microbicidal processes. The role of visible light as an antibacterial agent was pointed out by Angarano et al. [67]. They revealed that visible light in the high-energy range (400–420 nm) could inactivate biofilms formed by both Gram-negative *Pseudo-monas fluorescens* and Gram-positive *Staphylococcus epidermidisbacteria* [67]. It is hypothesized that the reactive oxygen species are involved in the antimicrobial effect of high-energy visible light [68]. Liou et al. assumed that the photocatalytic processes occurring on the surfaces of TiO_2_-based materials are the most important factors, which can facilitate the prevention of diseases caused by microorganisms [69]. These authors investigated the microbicidal activity of modified TiO_2_ systems (the modification consisted in shifting the absorption towards the visible light spectrum) against model bacteria *E. coli* and human pathogens showed that irradiation with visible light significantly decreased the number of test bacteria [69]. The results of the carried out and described here investigations showed that the PCL + *n*TOCs ((**1**), (**2**)) composites exhibit photocatalytic activity in the visible range by generating the reactive oxygen species (as was proved by EPR spectroscopy). Taking into account the received results we can assume that the ROS generated on the surface of the manufactured composite fittings under their influence of exposition on the visible light are the main antimicrobial factor.

## 4. Materials and Methods

### 4.1. Materials 

Titanium(IV) isobutoxide (Aldrich, St. Louis, MO, USA), 4-aminobenzoic acid (Aldrich, St. Louis, MO, USA), 9-fluorenecarboxylic acid (Organic Acros, Geel, Belgium) were purchased commercially and were used without further purification. All solvents used in synthesis, i.e., acetone and toluene were distilled before their use and stored in an argon atmosphere. The processes of Ti(IV) oxo-complexes synthesis were carried out using the standard Schlenk technique in the inert gas atmosphere (Ar) and at room temperature (RT).

### 4.2. Synthesis of Ti(IV) oxo-Complexes(TOCs) and Polymer/TOCs Composites

#### 4.2.1. The Synthesis of [Ti_4_O_2_(O^i^Bu)_10_(O_2_CPhNH_2_)_2_] (1)

The complex was synthesized, as reported [17]. 0.12 g of 4-aminobenzoic acid (0.875 mmol) was added to the solution of 1.19 g of titanium(IV) isobutoxide (3.5 mmol) in 2 ml of toluene, leading to clear yellow solution. The solution was left for crystallization. The yield basing on acid: 41% (0.22 g). Anal. Calc. for C_54_H_102_O_14_Ti_4_N_2_:C, 52.86; H, 8.38; N, 2.28; Ti, 15.61. Found: C, 53.14; H, 7.83; N, 2.05; Ti, 15.56. ^13^C NMR (solid state, 295 K, δ[ppm]): 14.3, 19.5 (CH_3_), 31.5, 34.7 (CH), 64.8, 78.6 (CH_2_), 113.6, 133.1 (C(Ph)), 152.4 (C-NH_2_), 175.4 (COO).

#### 4.2.2. The Synthesis of [Ti_4_O_2_(O^i^Bu)_10_(O_2_CC_13_H_9_)_2_] (2)

Complex was synthesized, as reported [17]. 0.184 g of 9-fluorenecarboxylic acid (0.875 mmol) was added to the solution of 1.19 g of titanium(IV) isobutoxide (3.5 mmol) in 2 ml of acetone, leading to clear yellow solution. The solution was left for crystallization. The yield basing on acid: 62% (0.36 g). Anal. Calc. for C_68_H_108_O_16_Ti_4_:C, 59.28; H, 7.93; Ti, 13.94. Found: C, 58.17; H, 7.86; Ti, 13.64. ^13^C NMR (solid state, 295 K, δ[ppm]): 14.6, 19.5 (CH_3_), 35.4 (CH), 55.6, 77.7 (CH_2_), 119.3, 127.5, 142.3 (C(Ph)), 178.4 (COO).

#### 4.2.3. The Composite Fittings Production

The poly(Ɛ-caprolactone) (PCL) fittings and the PCL composite ones, containing 5 and 20 wt.% of TOCs (Figure 9), were produced by using equipment for plastics processing consisting of a twin-screw extruder (Haake MiniLab II, Thermo Scientific, Waltham, MA, USA) and a pneumatic mini-injection molding machine (Haake MiniJet II, Thermo Scientific, Waltham, MA, USA). The following injection parameters were used: temperature of the plasticizing system: 70 °C; injection cylinder temperature: 80 °C; mold temperature: 25 °C; injection pressure: 400 bar; holding pressure: 300 bar; holding time: 3 s. The mechanical strength studies of the manufactured fittings in the static tensile test were carried out in the following conditions: tensile speed: 20 mm/min; jaw opening: 50 mm; length of the measuring section of the fittings: 40 mm; fittings thickness: 1.5 mm; measurement temperature: 25 °C. 

The compressive strength was determined using the same equipment. The rectangular samples’ dimensions were: width, 10 mm; thickness, 10 mm; and specimen gauge length, 3 mm. The condition of the test was: temperature, 25 °C; and compressive speed, 1.2 mm/min.

#### 4.2.4. The Composite Foils Production

Foils of composites based on poly(methyl methacrylate) (PMMA), i.e., PMMA + 20TOCs (TOCs = (**1**), (**2**)), have been prepared for comparison [17]. The TOCs powder portion (acting the 20 wt.% of the reaction mixture) was dispersed (with the use of ultrasonic bath) in the solution, which was received by dissolving 1 g PMMA in 5 cm^3^ of toluene. The composite films were produced by slow solvent evaporation in a glove box at room temperature.

### 4.3. Analytical Procedures

The structure of isolated solid synthesis products was confirmed using vibrational spectroscopy methods, i.e., IR spectrophotometry (Perkin Elmer Spectrum 2000 FT-IR spectrometer (400–4000 cm^−1^ range, KBr pellets)) and Raman microscopy (RamanMicro 200 spectrometer (PerkinElmer, Waltham, MA, USA)). Raman spectra were registered using a laser of wavelength 785 nm, with maximum power 350 mW, in the range 200–3200 cm^−1^, using a 20 × 0.40/FN22 objective lens and an exposure time of 15 s each time. Solid-state ^13^C NMR spectra were registered at 22 °C on Bruker AMX 300 (Cambridge Scientific Corp. Watertown, MA, USA). Elemental analyses were performed on an Elemental Analyser vario Macro CHN Elementar Analysensysteme GmbH (Hanau, Germany). 

The cross-sections of manufactured fittings were studied using a scanning electron microscope (SEM) with field emission and backscattered electron detector (BSE) (Quanta 3D FEG, Huston, TX, USA). The diffuse reflectance UV-Vis (UV-VIS-DRS) spectra registered between 200 and 800 nm were used for the absorption maximum determination. The Jasco V-750 spectrophotometer was used in our works (JASCO Deutschland GmbH, Pfungstadt, Germany). The electron paramagnetic resonance (EPR) spectroscopy was used in order to confirm the presence of reactive oxygen species on the surface of the investigated samples. Measurements were carried out using an X band EPR SE/X-2541M spectrometer (Radiopan, Poznań, Poland) with a 100 kHz modulation. The microwave frequency was monitored with a frequency meter. The magnetic field was measured with an automatic NMR-type JTM-147 magnetometer (Radiopan, Poznań, Poland). Measurement conditions: microwave frequency: ca. 9.33 GHz; microwave power: 20–40 mW; modulation amplitude: 0.2–1 mT; sweep: 10-50 mT; sweep time: 1–8 min.; time constant: 0.03–0.3 s; receiver gain: 3.2–5 × 10^5^. Low intensity signals were recorded in the accumulation mode. The measurements were performed for cut films of PCL + TOCs composites at room temperature. The thermal gravimetric analysis (TGA) and the differential scanning calorimetry (DSC) were performed using Thermal Analyzer STA 449 F5 Jupiter (Netzsch, Selb, Germany), the In, Bi, Zn, Al, and Au were used as the references. The analysis was carried out in the range 30–550 °C at a rate 5 °C/min in the nitrogen atmosphere. The mechanical strength studies of the polymer and composite fittings, subjected to the static tensile test (Model 4204, Instron, USA) were carried out in the following conditions: tensile speed: 20 mm/min; grip distance, 50 mm; length of the measuring section of the fittings, 40 mm; fittings thickness, 1.5 mm; measurement temperature, 25 °C.

### 4.4. The Photocatalytic Activity Evaluation of Produced (Polymer + nTOCs) Composites

The photocalatytic activity of PMMA + 20TOCs composites (TOCs = (1)–(2)) and PCL + *n*TOCs composites (TOCs = (1)–(2), *n* = 5, 20 wt.%) was studied by monitoring of MB aqueous solution decolorization. Samples of a 10 × 10 mm size were preconditioned by exposition to Vis light for 28 h. In the next steps, composites were placed in quartz cuvettes with dye solution (V = 3.5 cm^3^, *c* = 2.0 × 10^−5^ M). After 12 h in the dark, the solution was replaced by the final MB solution (V = 3.5 cm^3^, *c* = 1.0 × 10^−5^ M) intended for the kinetic measurements. The prepared samples were exposed to Vis light (77 W tungsten halogen lamp, range of 350–1200 nm). All cuvettes were covered with Teflon lid during irradiation. MB absorbance at 660 nm was registered (Metertech SP-830 PLUS, Metertech, Inc., Taipei, Taiwan) every 40 minutes for 30 hours of irradiation. Percentage of MB decolorization was calculated using the Equation (1):(1)$$\%\;\text{of dye decolorization}\;=\;[(c_{0}\;-\;c_{t})/c_{0}]\;\frac{c_{o}\;-\;c_{t}}{c_{o}}\;\times \;100$$
where *c*_0_ is an initial concentration of dye and *c*_t_ is a dye concentration at a given time *t* [42].

### 4.5. The Estimation of Antimicrobial Activity of Studied (Polymer + TOCs) Composites

Antibacterial and antifungal activities of PCL and PCL + TOCs composites (20 × 10 × 1 mm) was studied against Gram-positive (*Staphylococcus aureus* ATCC 6538 and *S. aureus* ATCC 25923) and Gram-negative (*Escherichia coli* ATCC 8739 and *E. coli* ATCC 25922) bacteria and yeasts of *Candida albicans* ATCC 10231. All strains were purchased from the American Type Culture Collection (Manassas, VA 20110 USA). Tested samples were sterilized in 70% ethanol for 15 minutes and then using a UVC lamp for 15 min in the laminar hood (Bioquell, Hampshire, UK) each side. Specimens were subsequently treated with indoor light (visible light). The earlier investigations revealed that the absorption maximum of PCL + TOCs samples was found at the UVA-Vis border, which enables the use of UVC radiation for their sterilization. Sterile samples were placed in sterile 25 ml screw-cap tubes (Sarstedt) with 2 ml of tested microorganism suspension (1.0–5.8 × 10^6^ colony forming units (c.f.u) cm^3^/^−1^) prepared in sterile deionized water and incubated for 24 h at 37 °C in a humid atmosphere and gently shaken (80 rpm) conditions. Samples were then vortexed and specimens removed from the tubes. Microbial suspensions were then serially ten-fold diluted and aliquots (100 µl) of each dilution were aseptically spread over the surface of Triptic Soy Agar (TSA, Becton Dickinson, USA) for bacteria and Sabouraud Dextrose Agar (SDA, Becton Dickinson) for yeasts poured into Petri plates. Inoculated plates were incubated at 37 °C for 24 h. Assays were performed in triplicate. After incubation, colony-forming units (cfu) were counted on the agar plates. Control was the suspension of test microorganisms in the tube. 

The antimicrobial activity was determined on the basis of the reduction factor (R) calculated according to the Equation (2):*R* = *U*_*t*_ − *A*_*t*_(2)
where *U_t_* is the average of the common logarithm of the number of viable bacteria on Petri plate, recovered from the untreated microbial suspension after 24 h, *A_t_* is the average of the common logarithm of the number of viable bacteria, on Petri plate, recovered from the treated microbial suspension with PCL or PCL + TOCs after 24 h. *R* ≥ 2 determines biocidal activity (at least 99% reduction of microbial growth).

## 5. Conclusions

The studies carried out lead to the production of PCL + *n*TOCs (*n* = 5 and 20 wt%) composites and to the determination of their physicochemical properties. Using the equipment for plastics processing, the tetranuclear Ti(IV)-oxo complexes (TOCs), of the general formula [Ti_4_O_2_(O^i^Bu)_10_(O_2_CR’)_2_] (R’ = PhNH_2_ (**1**) and C_13_H_9_ (**2**)), were dispersed in the poly(ε-caprolactone) (PCL) matrix. The enrichment of the polymer with micro-grains of TOCs slightly affects PCL thermal and mechanical properties while significantly shifting the absorption maximum towards the visible range.

The photocatalytic activity of produced (polymer + TOCs) composites were assessed using the methylene blue (MB) photoinduced decolorization process in the visible range. The obtained results revealed the clear photocatalytic activity of composites containing 20 wt.% of (**1**) and (**2**). Analysis of EPR spectra of the above-mentioned samples proved that photo-excitation of their surface by visible light leads to ROS generation. According to EPR data, the paramagnetic species of O^−^ were generated on the surface of PCL + 20(**1**) composite sample while both O^−^ and O_2_^−^ ones on the surface of PCL + 20(**2**). The obtained results suggest that the type of carboxylic ligands can influence the type of appearing ROS specimens.

From the studied composite samples, the strong antibacterial activity (against *E. coli* and *S. aureus* strains) and weak inhibition of yeasts (*C. albicans*) growth was found for composites PCL + 20 TOCs (TOCs = (**1**) and (**2**)). However, good antibacterial properties also revealed the PCL + 5(**2**) system. The obtained results indicate that the composite containing the [Ti_4_O_2_(O^i^Bu)_10_(O_2_CC_13_H_9_)_2_] micro grains exhibit better bactericidal activity, which can be associated with the ability to generate more differentiated forms of ROS (O^−^, O_2_^−^) during exposure to visible light, suggesting that the produced materials can be considered as the materials with self-disinfecting activity. It should be noted that this paper includes the preliminary results of studies on the antimicrobial potential of TOCs dispersed in the PCL matrix. A detailed analysis determining the mechanisms of antimicrobial action of polymer + TOCs composites will be carried out during the next step of our works and will form the basis of another publication.

## Figures and Tables

**Figure 1 ijms-22-07021-f001:**
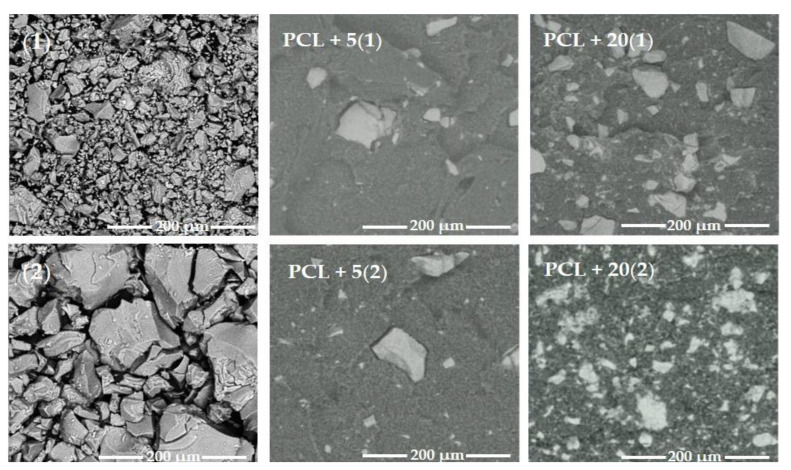
SEM images of isolated micro-powders of oxo-complexes (**1**) and (**2**) and cross sections of the produced PCL + *n*TOCs (TOCs = (**1**) and (**2)**) composite fittings.

**Figure 2 ijms-22-07021-f002:**
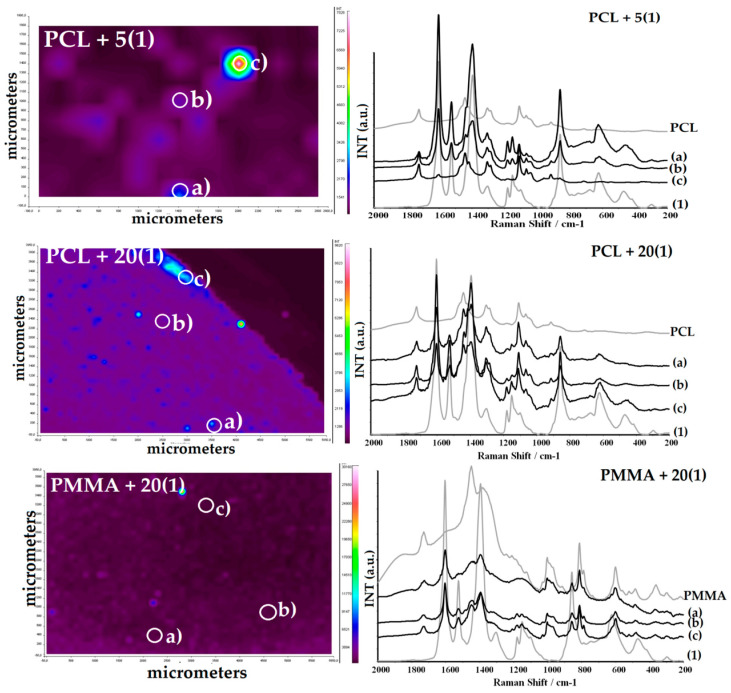
Raman microscopy maps and spectra of PCL + 5(**1**), PCL + 20(**1**), and PMMA + 20(**1**) composites.

**Figure 3 ijms-22-07021-f003:**
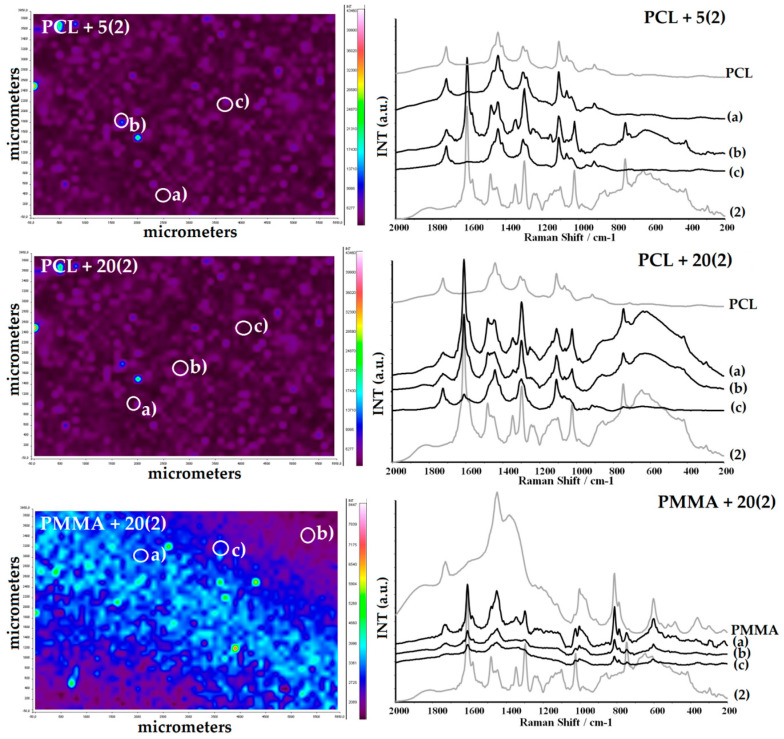
Raman microscopy maps and spectra of PCL + 5(**2**), PCL + 20(**2**), and PMMA + 20(**2**) composites.

**Figure 4 ijms-22-07021-f004:**
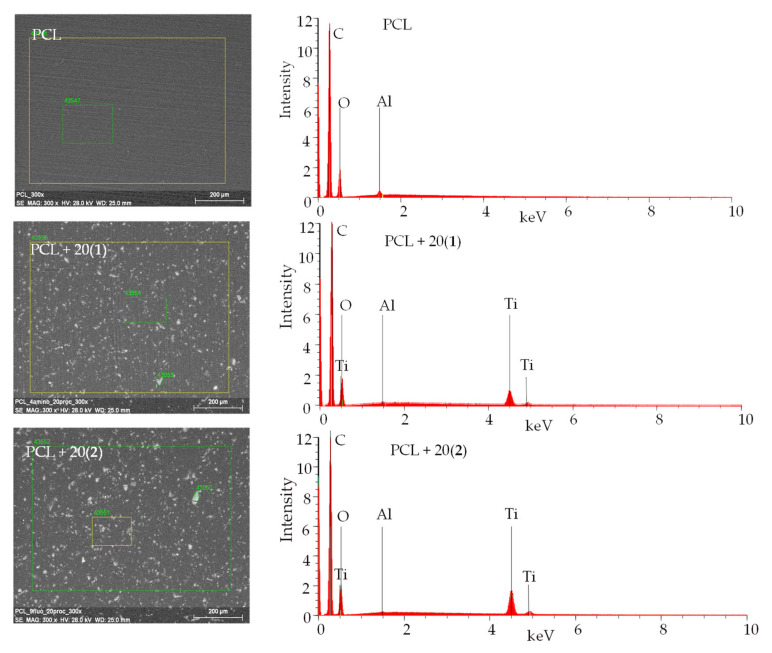
SEM images and EDX patterns of (PCL + 20(**1**)) and (PCL + 20(**2**)) composites, the pure PCL spectra are given for comparison.

**Figure 5 ijms-22-07021-f005:**
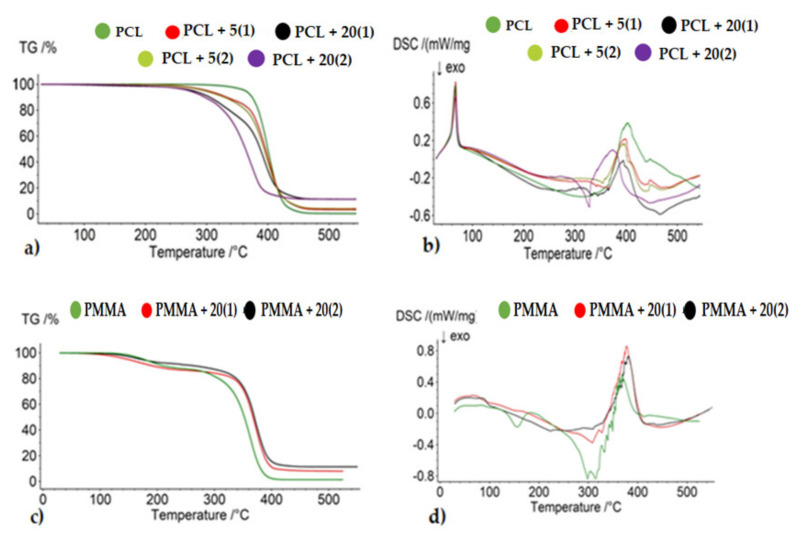
Thermogravimetric curves (TGA) (**a**,**c**) and the differential scanning calorimetry curves of (DSC) (**b**,**d**) of the produced composite materials.

**Figure 6 ijms-22-07021-f006:**
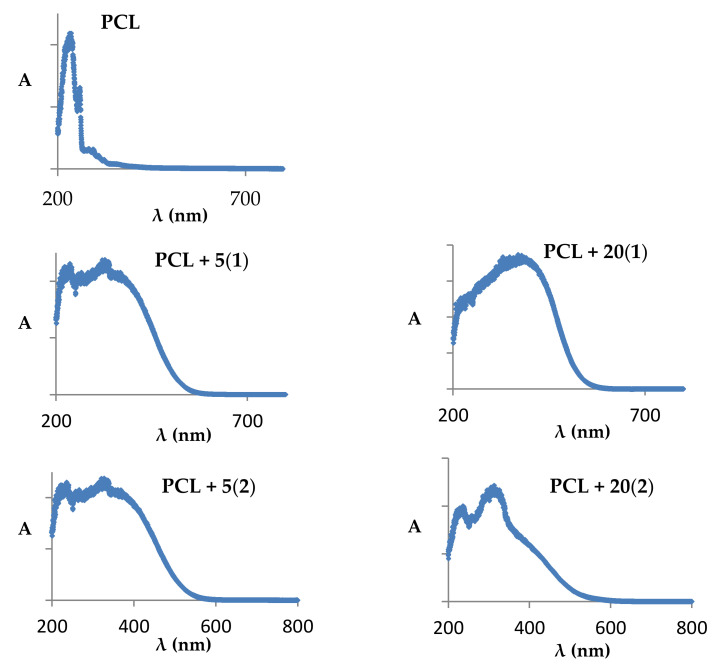
UV–Vis–DRS spectra of PCL and studied PCL + nTOCs composites.

**Figure 7 ijms-22-07021-f007:**
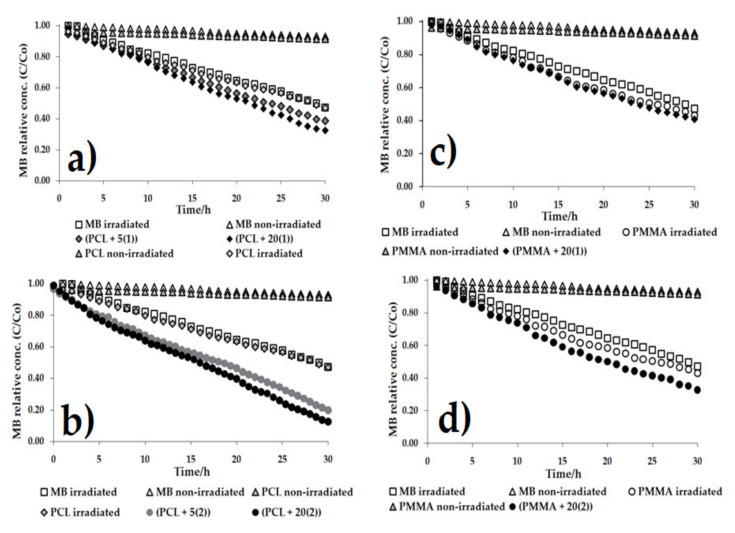
Changes in the concentrations of the methylene blue (MB) solution as a function of time for the respective composite materials irradiated with VIS radiation ((**a**) PCl + *n*(**1**), (**b**) PCL + *n*(**2**) (*n* = 5, 20 wt.%), (**c**) PMMA + 20(**1**), (**d**) PMMA + 20(**2**) (*n =* 20 wt.%).

**Figure 8 ijms-22-07021-f008:**
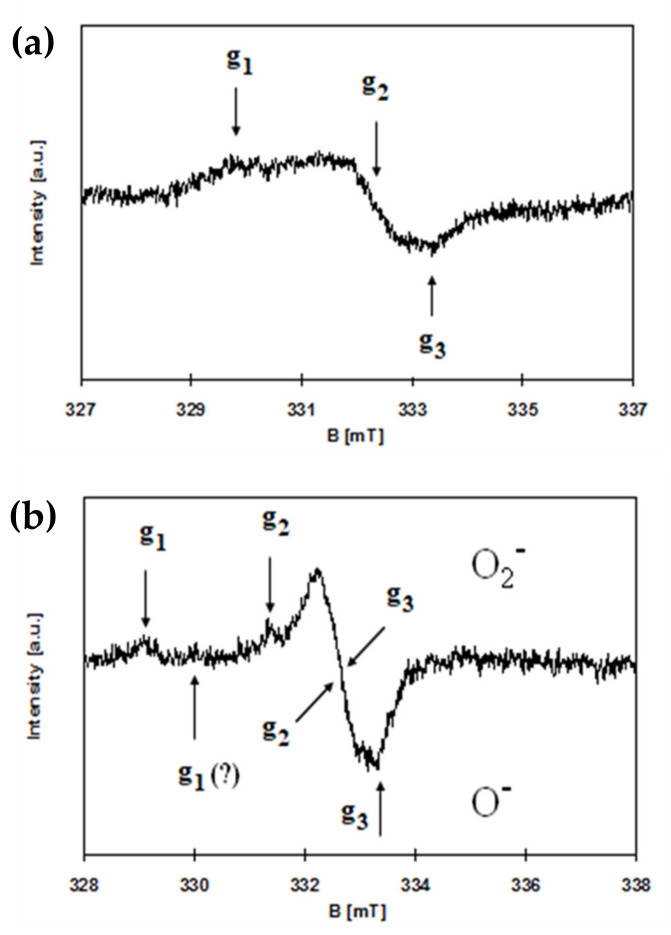
EPR spectra of PCL + 20(**1**) (**a**) and PCL + 20(**2**) (**b**).

**Figure 9 ijms-22-07021-f009:**
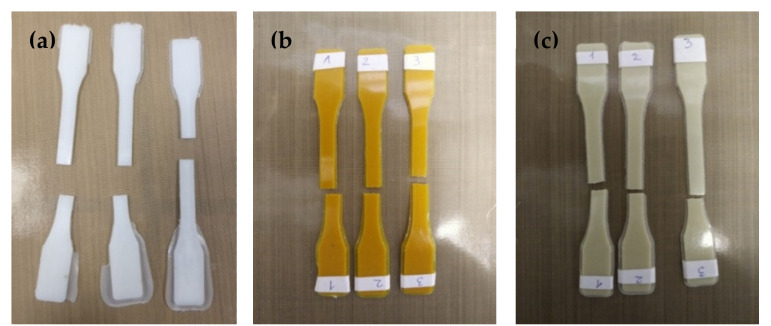
Fittings produced by injection molding method, samples after breaking on a testing machine: (**a**) PCL, (**b**) PCL + 20(1), and (**c**) PCL + 20(2).

**Table 1 ijms-22-07021-t001:** SEM EDX quantitative data; mean value of mass percent (%).

Composite	C	O	Al	Ti
PCL	28.35	70.98	0.67	-
(PCL + 5%(**1**))	26.15	66.53	0.32	7.00
(PCL + 20%(**1**))	25.23	65.73	0.21	8.83
(PCL + 5%(**2**))	26.06	66.15	0.37	7.42
(PCL + 20%(**2**))	22.79	65.39	0.20	11.62

**Table 2 ijms-22-07021-t002:** Thermal parameters received from thermogravimetric analysis (TGA) and differential scanning calorimetry (DSC) of the composites (T_g_ = glass transition temperature, T = evaporation temperature of an unreacted monomer, T_m_ = melting temperature, T_d/max_ = decomposition temperature (transition maximum), T_max_ = temperature in the thermal transition maximum, Δm = thermal transition weight loss).

	DSC					TGA		
Composite	T_g_/^o^C	T_m_/^o^C	T/^o^C	T_d_^o^C	T_d/mx_/^o^C	Stage I	Stage II	Solid Residue
						T_max_/^o^C/Δm/%	T_max_/^o^C/Δm/%	%
PCL	-	66.1	-	-	402.3	400.1/100	-	0
(PCL + 5(**1**))	-	66.7	-	-	398.3	397.9/97	-	3
(PCL + 20(**1**))	-	65.5	-	-	394.9	394.7/89	-	11
(PCL + 5(**2**))	-	66.7	-	353.6	395.5	396.3/96	-	4
(PCL + 20(**2**))	-	67.2	-	328.0	373.6	366.2/89	-	11
PMMA	99.6	-	150.8	-	368.6	199.9/12	365.1/85	3
(PMMA + 20(**1**))	99.0	-	-	-	377.8	159.9/15	372.7/76	9
(PMMA + 20(**2**))	100.9	-	-	-	381.3	189.8/12	371.0/76	12

**Table 3 ijms-22-07021-t003:** The results of composite fittings mechanical strength and compressive strength studies in the static tensile test.

Sample	E_tensile_ [MPa]	σ_max_ [MPa]	Ɛ_max_ [%]	σ_BR_ [MPa]	Ɛ_BR_ [%]	E_compressive_ [MPa]	σ_yield_ [MPa]	Ɛ_yield_ [%]
PCL	422 + 13	21.0 + 1.4	6.6 + 0.9	20.7 + 1.3	6.8 + 0.9	100 + 4	8.4 + 0.1	20.0 + 0.6
PCL + 20(**1**)	504 + 21	15.9 + 0.6	4.2 + 0.1	15.9 + 0.6	4.2 + 0.2	107 + 3	8.4 + 0.1	20.1 + 0.3
PCL + 20(**2**)	480 + 3	15.8 + 0.5	4.5 + 0.1	15.7 + 0.5	4.5 + 0.1	115 + 1	8.5 + 0.1	19.1 + 0.5

E,Young’s modulus (MPa); σ_max,_ maximum stress (tensile strength) (MPa); Ɛ_max_, strain at the maximum stress (%); σ_BR_, stress at break (MPa); Ɛ_BR_, strain at break (%); E_compressive_, compressive strength (MPa); σ_yield_, stress at the yield (MPa); Ɛ_yield_, strain at the yield (%).

**Table 4 ijms-22-07021-t004:** MB solution decolorization percentages and ΔA_180_ parameters for the studied reactions in relation to the composites.

**Composite**	**MB Decolorization ^a^ (%)**	**ΔA 30**	**ΔA 30** **in Reference** **to MB**	**10^2^ Rate Constant** **h^−1^**	**10^2^ Rate Constant in Reference to MB,** **h^−1^**
MB irradiated	76.35	0.749	-	2.50 + 0.02	-
PCL	76.91	0.726	−0.023	2.51 ± 0.03	0.01
PCL + 5(**1**)	75.54	0.701	−0.048	2.79 ± 0.04	0.29
PCL + 20(**1**)	81.46	0.782	0.033	2.96 ± 0.05	0.46
PCL + 5(**2**)	79.56	0.755	0.006	3.28 ± 0.04	0.78
PCL + 20(**2**)	87.32	0.847	0.980	3.40 ± 0.06	0.90
**Composite**	**MB Decolorization ^a^** **(%)**	**ΔA 30**	**ΔA 30** **in Reference** **to MB**	**10^2^ Rate Constant h^−1^**	**10^2^ Rate Constant in Reference to MB,** **h^−1^**
MB irradiated	76.35	0.749	-	2.50 ± 0.02	-
PMMA	75.18	0.751	0.002	2.60 ± 0.03	0.10
PMMA + 20(**1**)	83.75	0.804	0.055	2.63 ± 0.04	0.13
PMMA + 20(**2**)	85.20	0.787	0.038	2.74 ± 0.07	0.24

Methylene blue (MB) decolorization at the end of the measurements (t = 30 h).

**Table 5 ijms-22-07021-t005:** EPR data for PCL + 20TOCs (TOCs = (**1**) and (**2**)) composite samples. The samples were exposed to visible light prior to measurement.

Sample	g-Factors	Species
PCL + 20(**1**)	2.020, 2.005, 1.999	O^−^
PCL + 20(**2**)	2.025, 2.010, 2.003(?), 2.019, 2.003, 1.999	O_2_^−^ O^−^

**Table 6 ijms-22-07021-t006:** Antimicrobial activity of PCL and PCL + *n*TOCs.

	Microorganisms				
Sample	*Escherichia coli* ATCC 8739	*Escherichia coli* ATCC 25922	*Staphylococcus aureus* ATCC 6538	*Staphylococcus aureus* ATCC 25923	*Candida albicans* ATCC 10231
	R	R	R	R	R
PCL	0.9	0.9	0.1	0.6	0.0
PCL + 5(**1**)	1.6	0.7	0.2	0.7	0.3
PCL + 5(**2**)	2.0	2.6	1.1	2.3	0.3
PCL + 20(**1**)	2.0	2.0	2.0	2.5	0.7
PCL + 20(**2**)	4.5	3.5	2.0	2.6	0.9

R, reduction index. R ≥2 is a biocidal effect when the microbial growth is reduced at least 100 times (99.0%).

## Data Availability

Data is contained within the article or Appendix A.

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
