# Peer review of "The Composites of PCL and Tetranuclear Titanium(IV)-oxo Complexes as Materials Exhibiting the Photocatalytic and the Antimicrobial Activity"

_ijms, 2021, doi:10.3390/ijms22137021_

Round 1

Reviewer 1 Report

This MS describes photocatalytic properties of tetranuclear titanium(IV) oxo complexes dispersed in polymeric substances. The study is well conducted and the results clearly analyzed. I have just a minor remark concerning the occurence of unmodified tetranuclear complexes in such materials. Why authors have not made 13-C solid-state NMR that is a crucial technique for identifying molecular species in solution or in crystals? This is the main weak point of the MS. Before publishing the MS, authors should thus address this point, either by providing solid-state 13-C NMR or by clearly stating that they plan to do such crucial measurements in a very next future. Otherwise, this is a good piece of work that could be of interest for the community working on photocatalytic properties of Ti(IV)-based compounds. I thus recommend publication in IJMS after minor revisions along the lines indicated above.

Author Response

Thank you for your valuable attention. In our earlier researches, we have focused mainly on determining the structure of TOCs, especially the {TiaOb} core architecture. For this purpose, X-ray diffraction, vibrational spectroscopy (IR, Raman), UV-VIS spectrophotometry, and mass spectrometry methods were used. However, the 13C NMR spectra in the solid of the isolated oxo-complexes were also registered. According to the reviewer's remark, in the experimental part of this work, the information on chemical shifts of peaks recorded in 13C NMR spectra of TOCs (1) and (2) has been also added. In our further investigations, a detailed analysis of NMR spectra of synthesized oxo-complexes in the solid and the solution will be carried out to determine the relationship between the core architecture of {TiaOb} and the chemical shifts of signals attributed to coordinated organic ligands.

Reviewer 2 Report

This paper reports the synthesis of composite materials produced by the dispersion of titanium(IV)-oxo complexes (TOCs) in poly(ε-caprolactone) (PCL) matrix. The primary reason for doing this was to develop antimicrobial materials. The research adds new knowledge to the literature, however, the study requires some more exploration. In the following, I will describe the shortcomings and requirements to fulfill them - 

  1. Even though the study is about composites, there should be some data on the characterization of the TOCs in this paper. For instance, SEM or TEM images are highly required.
  2. If the primary point was to make these antibacterial composites, the study should have thorough antibacterial studies. Just one assay would not suffice. The photocatalytic processes occurring on the surfaces of TiO2-based materials might not be the only mechanism. 
  3. The labelling of the SEM images in Figure 1 needs to be clearer. 
  4. For composites like these, it is important to study the compressive strength. Only tensile tests are not enough. Specifically, the stiffness changes should be explored. 

Author Response

In the revised version of our manuscript, the shortcomings and requirements suggested by the reviewer have been considered. 

1. "Even though the study is about composites, there should be some data on the characterization of the TOCs in this paper. For instance, SEM or TEM images are highly required."

According to the reviewer remark, the SEM images of TOCs micro-powders ((1) and (2)) are presented in Figure 1. Moreover, SEM-EDX data was added as the supplementary data (Figure S1). It should be noted that the detailed data concerning TOCs (1) and (2) were presented in our eariler papers [17, 30].

2. "If the primary point was to make these antibacterial composites, the study should have thorough antibacterial studies. Just one assay would not suffice. The photocatalytic processes occurring on the surfaces of TiO2-based materials might not be the only mechanism."

We agree with the reviewer point of view and are thankful for this nice remark. This paper includes preliminary studies on antimicrobial potential of test composites. Advanced analyses determining the mechanisms of antimicrobial action of such composites will be carried out in the near future and will include determination of cell membrane damage/cell viability (crystal violet assays, enzyme release e.g. cell hydrolases) as well as metabolic activity in microbial cells based on dehydrogenases or ATP content. The microbial growth kinetic will be also performed when cells exposed for such antimicrobial  surfaces.

Taking into account the number of analysed strains and proposed analyses the obtained results will form the basis of another publication.

3. "The labelling of the SEM images in Figure 1 needs to be clearer."

According to reviewer remark, the labelling’s in Figure 1 have been changed.

4. "For composites like these, it is important to study the compressive strength. Only tensile tests are not enough. Specifically, the stiffness changes should be explored."

According to reviewer suggestion, the Compressive strength of samples were determined.

Round 2

Reviewer 2 Report

The authors have addressed all the comments.